# Impact of Green Innovation Efficiency on Carbon Peak: Carbon Neutralization under Environmental Governance Constraints

**DOI:** 10.3390/ijerph191610245

**Published:** 2022-08-18

**Authors:** Meng Guo, Shukai Cai

**Affiliations:** School of Economics and Management, Anhui Polytechnic University, Wuhu 241000, China

**Keywords:** carbon dioxide emissions, green innovation efficiency, carbon peak, carbon neutralization

## Abstract

Under environmental governance constraints, in order to explore the quantitative contribution of green innovation efficiency to carbon peak and carbon neutralization at the urban level, this paper uses the unexpected Super-SBM model to measure the green innovation efficiency of each prefecture-level city based on the panel data of 40 prefecture-level cities in the Yangtze River Delta from 2010 to 2019. Furthermore, the panel fixed effect model is constructed, and the two-stage least squares estimation method is used for empirical research. It is found that green innovation efficiency can significantly reduce carbon emissions in the Yangtze River Delta, promote carbon emissions in the Yangtze River Delta to reach an early peak, and achieve the long-term goal of carbon neutrality as soon as possible. This conclusion is still stable after solving the endogenous problem and the influence of outliers. The results of regional heterogeneity analysis show that green innovation efficiency has remarkable effects on carbon emission reduction in Anhui and Zhejiang Provinces, and the emission reduction effect in Zhejiang Province is greater than that in Anhui Province. In addition, there exists obvious heterogeneity between different quantiles for the impact of green innovation efficiency on carbon emissions, showing an “inverted U” shape, and its intensity in the context of medium carbon emissions is greater than that of low carbon and high carbon emissions.

## 1. Introduction

A large amount of evidence shows that global warming caused by anthropogenic greenhouse gas emissions (GHGs) has become one of the main challenges to human welfare [1,2,3]. Under the background of the new era, global warming caused by carbon emissions has been paid more and more attention by the international community [4]. Addressing climate change has become a common challenge facing the world [5]. Under the background of economic and social development, urbanization and the increase in energy consumption, China’s carbon emissions have increased rapidly in the past two decades. As the world’s largest developing country, China is at a critical stage of industrialization and urbanization and faces more severe challenges in coordinating economic growth and reducing carbon emissions. In 2007, China’s total CO_2_ emission exceeded that of the United States, ranking the first in the world. While in 2013, China accounted for 28% of global carbon emissions, and per capita emissions exceeded those of the European Union for the first time [5]. By 2020, China’s total carbon emission reached 9.899 billion tons, still ranking the first in the world, and its share of the world’s carbon emission increased to 31%, whose carbon emission share in the world’s major countries is still growing.

In this regard, the Chinese government has put forward a series of emission reduction measures. At the Copenhagen climate conference, China pledged to peak its carbon dioxide emissions by 2030 and strive to achieve carbon neutrality before 2060. In order to achieve the strategic goal of carbon neutrality, China must change the mode of economic development and take the green low carbon development path of energy saving and emission reduction. That is, by changing the pattern of economic development and improving production efficiency, taking economic growth and energy conservation and emission reduction into account, we can realize green, low-carbon and high-quality development. In this case, as an important factor to achieve a win-win goal of environmental protection and technological progress, green innovation has become an inevitable choice for regions to win competitive advantages and achieve economic development under increasingly stringent environmental regulations [6]. Green innovation efficiency has a catalytic effect on regional economic growth and further affects the level of economic growth of the whole country [7]. Therefore, in the current practice of China’s national strategy of higher quality integrated development of the Yangtze River Delta, it is more important to adhere to the concept of “innovation” and “green” development, take the road of “ecological priority, and green development”, and improve the efficiency of green innovation [8].

The concept of ‘green innovation’ was first proposed by Fussler et al. [9] in 1996. Green innovation is considered to be an effective way to reduce pollution, reduce energy consumption, save energy and achieve sustainable economic growth [10,11]. Its essence refers to the activities with commercial value carried out by enterprises to reduce environmental pollution. Green innovation efficiency refers to the performance in the development of green innovation. It is generally believed that green innovation efficiency includes environmental benefits in innovation input and output, and obtains the optimal innovation output at the lowest cost of resources and environment. The improvement of green innovation efficiency means the progress of technology, which can reduce carbon dioxide emissions at the source while maintaining economic growth [12]—that is, green innovation efficiency can effectively reduce carbon dioxide emissions and promote the realization of the strategic goal of carbon peak and carbon neutralization in theory.

At present, the research on carbon peak and carbon neutralization in the Yangtze River Delta is mostly based on the LMDI index decomposition method to explore the driving factors of carbon emissions and the scenario prediction method to analyze the peak time under different carbon emission scenarios. It is rare to discuss its quantitative contribution to the strategic goal of carbon peaking and carbon neutralization in the Yangtze River Delta from the perspective of green innovation efficiency. So, how does the efficiency of green innovation affect the strategic goal of carbon peak and carbon neutrality in the Yangtze River Delta? In this context, to answer this question, this paper uses the Unexpected Super SBM model to measure the green innovation efficiency of various prefecture-level cities in the Yangtze River Delta. On this basis, a fixed effect model is constructed to quantitatively analyze the quantitative contribution of green innovation efficiency to carbon peak and carbon neutralization. On this basis, this paper constructs a fixed effect model to quantitatively analyze the quantitative contribution of green innovation efficiency to carbon peak and carbon neutralization, and solves endogenous problems with instrumental variables. Additionally, this paper explores regional heterogeneity by grouping regression at the provincial level, and uses the panel quantile regression to explore the heterogeneity of the effect of green innovation efficiency on carbon emission reduction under different carbon emission intensities. Finally, this paper puts forward corresponding policy suggestions to promote the Yangtze River Delta region to achieve the long-term goal of carbon peak and carbon neutrality as soon as possible from the perspective of green innovation efficiency.

The other parts of this paper are organized as follows: Section 2 is the literature review of related topics. Section 3 is the research hypothesis. Section 4 introduces the research methods and the measurement of important indicators. Section 5 is the description of the current situation of carbon emissions in the Yangtze River Delta. Section 6 is empirical analysis, including benchmark regression analysis, endogenous test, robustness test and heterogeneity analysis. Section 7 is the policy recommendations and the shortcomings of this paper.

## 2. Literature Review

### 2.1. Foreign Literatures Review

In a literature review of foreign studies, it is found that research on carbon emission reduction has a long history and has formed a large number of theoretical and practical results. These studies mainly focus on the influencing factors of carbon emissions, energy consumption and carbon emissions trajectory.

The influencing factors of carbon emissions mainly include population size, economic development, per capita income and low-carbon technology [13,14]. Specifically, Rocío Román-Collado et al. and Boqiang Lin and Izhar Ahmad both explored the impact of population on carbon emissions. The former analyzed the decoupling elasticity and two-stage decomposition of energy consumption in Colombia from 2000 to 2015. Analytical results show that population and activity effects contribute to increasing energy consumption in the country, while intensity effects and, to a lesser extent, structural effects contribute to reducing energy consumption [15]. The latter decomposed the extended Kaya identity by the log mean decomposition index (LMDI) decomposition model and clarified that the population growth is the main factor for increasing energy-related carbon dioxide emissions [16]. On the contrary, Rawshan et al. took Malaysia as an example and used the ARDL boundary test method. They found that per capita energy consumption and per capita GDP have a long-term positive impact on per capita carbon emissions during the study period, but the population growth rate has no significant indigenous impact on per capita carbon dioxide emissions [17]. It can be seen that the effect of population effect on carbon emissions is heterogeneous in different countries. In the research on the effect of economic growth on carbon emission reduction, Abid and Mehdi found that there is a monotonic increasing relationship between carbon dioxide emissions and total GDP (the sum of formal economy and informal economy) in the presence of informal economy [18]. Jungho and Baek used time series data from individual countries to test the Environmental Kuznets Curve (EKC) hypothesis. The ARDL method was used to evaluate the impact of per capita income of Arctic countries on carbon dioxide emissions [19]. In the study of the impact of income on carbon emissions, Cenjie Liu et al. considered the short-term and long-term impact of income inequality on carbon emissions; they used the panel ARDL model and quantile model to analyze the impact of income inequality on carbon emissions in American states. Research showed that income inequality increases US carbon emissions in the short term but increases US carbon emissions in the long term [20]. 

Some scholars have conducted research at the enterprise or industrial level. Taking the enterprises in South American countries as the research object, Carmen Córdova et al. explores the impact of company size, assets and financial status at the enterprise level on whether to disclose carbon emissions and its evolution with the help of Logit and the linear panel data model [21]. Jeong and Kim looked at changes in CO_2_ emissions of Korean industrial manufacturing in 1991 and 2009 from the perspective of multiplication and addition. They found that the intensity effect and structural effect have significant positive effects on carbon dioxide emission reduction in South Korea [22]. On this basis, some scholars forecasted carbon emissions in different scenarios. The prediction of energy consumption and carbon emission trajectory is mainly based on non-numerical simulation by artificial intelligence software, such as fuzzy logic, genetic algorithm, neural network, support vector machine, ant colony algorithm, and particle swarm optimization algorithm. Specifically, Uzlu et al. predicted energy consumption in Turkey using artificial neural networks [23]. Vaillancourt et al. built a multi-regional TIMES-Canada model to calculate energy consumption trends in Canada by 2050. The results showed that energy consumption in Canada will increase by 43 percent in 2050 compared with 2007 levels [24]. Ram M. Shrestha and Salony Rajbhandari took Kathmandu Valley in Nepal as an example to explore the impact of three carbon emission reduction targets on energy and environment in the baseline scenario. The research results showed that in order to achieve the goal of 30% cumulative CO_2_ emission reduction (ER30), a major shift in energy use patterns from oil and gas to electricity is required [25]. 

### 2.2. Chinese Literature Review

Combing the relevant literature in China, scholars have studied the trajectory characteristics, driving factors, emission reduction strategies and peak path of carbon emissions from different spatial scales. 

Early scholars focused on the trajectory characteristics and important influencing factors of carbon emissions. Initially, scholars explored the decoupling between carbon emissions and economic growth using the Tapio decoupling elasticity coefficient [26,27,28]. Wang Min’s research results showed that the changes of energy consumption and carbon emissions in Qinghai Province are basically decoupled from the growth of economic aggregate [29]. Zhang Youguo and Bai Yujie analyzed the carbon decoupling index of China’s provinces from the ‘Ninth Five-Year’ period to the ‘Thirteenth Five-Year’ period. The results showed that the economic development of most provinces continued to present strong or weak decoupling by the first half of the Thirteenth Five-Year. [27]. In addition, scholars mainly used the LMDI model to identify important influencing factors of carbon emissions. Yue Shujing decomposed the influencing factors of carbon emissions in the Yangtze River Delta urban agglomeration into population size, per capita output, industrial structure, and industrial carbon intensity by LMDI index decomposition [29]. Peng Song and Huimin Zhang built the localized LEAP model and determined that the industrial terminal energy intensity, energy consumption structure, industrial structure and power production structure are the key factors affecting the carbon peak goal of Chongqing city with the help of LMDI decomposition and the Tapio decoupling elasticity coefficient [26]. Wang and Feng also explored the green and low-carbon development of Qinghai Province based on the LMDI model and decoupling index and clarified that population and economic growth are the main driving forces of carbon emissions in Qinghai Province [30]. 

On this basis, scholars predicted the time of carbon peak and carbon neutralization through scenario simulation and proposed differentiated peak paths. Such studies mainly focus on provincial and urban levels. Guo Fang and Wang Can used the Monte-Carlo method and K-means clustering algorithm to construct the index system to divide 286 cities in China into five categories: low-carbon potential cities, low-carbon demonstration cities, population loss cities, resource-dependent cities, and traditional industrial transition cities [30]. In view of different types of cities, this paper puts forward practical suggestions on the goal design and action focus of urban carbon peak. Based on the carbon emission characteristics of 31 provinces, Zhang and Li divided the 31 provinces into five categories based on the heterogeneity of economic development, industrial structure, energy consumption, and emission characteristics by the hierarchical clustering method. Additionally, they put forward differentiated approaches based on the progress of provincial peak action [31]. Kai Fang et al. developed an extended STIRPAT model to study whether the future energy-related emissions of 30 provinces in China will reach the peak and how to reach the peak. The prediction results were integrated into the scenario analysis to simulate, and the time range and peak range of China’ s carbon emissions were clarified [32]. 

In addition, some scholars considered the liquidity characteristics of carbon emissions to explore the linkage effect of regional collaborative emission reduction. Tan and Jiang used the Topsis method and gray correlation theory to comprehensively measure the coordination level of inter-provincial carbon emission reduction intensity in China from 2011 to 2019. They used the Gini coefficient, δ-convergence model, and the β-convergence model to analyze the unbalanced development trend of carbon emission reduction in China. They found that optimizing regional economic structure is conducive to promoting regional coordination of carbon emission reduction [33].

To sum up, many scholars have conducted in-depth research on issues related to China’s realization of carbon peaking and carbon neutralization from a variety of perspectives, which provides a good reference for the early realization of the strategic objectives of carbon peak and carbon neutralization. However, the current research on carbon emissions in the Yangtze River Delta pays more attention to the current situation and peak path of carbon emissions, and the exploration of influencing factors is mostly based on the exponential decomposition method, rarely from the perspective of green innovation efficiency. Therefore, this paper selects the Yangtze River Delta urban agglomeration with developed economy, education, science, and technology to explore the quantitative impact of green innovation efficiency on carbon peak and carbon neutrality of the Yangtze River Delta urban agglomeration from the urban level. It is expected to provide relevant reference for the goal design of carbon peaking and the realization path of carbon neutralization in the Yangtze River Delta urban agglomeration. The possible marginal contributions of this paper are: This paper quantitatively analyzes the contribution of carbon peak and carbon neutralization of Yangtze River Delta urban agglomeration from the perspective of green innovation efficiency, enriches the research at the level of urban agglomeration, and provides relevant reference for the goal design of carbon peak and the realization path of carbon neutralization of Yangtze River Delta urban agglomeration.

## 3. Research Hypothesis

Firstly, compared with traditional innovation, green innovation has ‘double externalities’, which can achieve the ‘win-win’ of economic development and environmental benefits. Green innovation is considered an effective way to reduce pollution, reduce energy consumption, save energy and achieve sustained economic growth. Secondly, while maintaining high-quality economic development, it can take into account environmental benefits, achieve technological progress, promote the transformation and upgrading of industrial structure, and reduce carbon dioxide emissions by improving the efficiency of green innovation. Improving green innovation efficiency means technological progress, reducing carbon dioxide emissions from the source and achieving optimal innovation output at the lowest environmental cost. Improving environmental performance, enhancing competitiveness and upgrading industrial structure through green innovation has become a common practice around the world. Urban innovation activities promote the flow of R&D capital and funds to sectors with higher profits, and contribute to the concentration of green innovation resources, thereby promoting the improvement of industrial green technology, the improvement of output quality and the agglomeration of green innovation resources. The proportion of the tertiary industry with high value-added and low energy consumption will be increased. The market share of primary and secondary industries with large pollution, large energy consumption and less value creation may gradually decrease, thus optimizing the industrial structure. Therefore, the improvement of green innovation efficiency can effectively promote the reduction in carbon dioxide emissions in theory. Based on the above analysis, Hypothesis 1 of this paper is proposed:

**Hypothesis** **H1:** 
*Green innovation efficiency can reduce carbon dioxide emissions and promote the Yangtze River Delta region to achieve the long-term goal of carbon peak and carbon neutralization as soon as possible.*


There are great differences in economic development, technical level, environmental status and resource endowment among cities in the Yangtze River Delta; thus, there may be regional heterogeneity in the emission reduction effect of green innovation efficiency. In addition, carbon emissions and the peak progress of each prefecture-level city in the Yangtze River Delta are not the same, and innovation atmosphere and motivation are also significantly different. The green innovation efficiency of Yangtze River Delta urban agglomeration is unevenly distributed in space. Is the effect of green innovation efficiency on carbon emissions heterogeneous under different carbon emission distributions? Based on the above analysis, Hypotheses 2 and Hypotheses 3 of this paper are proposed:

**Hypothesis** **H2:** 
*The effect of green innovation efficiency on reducing carbon dioxide emissions has regional heterogeneity.*


**Hypothesis** **H3:** 
*The effect of green innovation efficiency on reducing carbon dioxide emissions is heterogeneous at different carbon emission quantiles.*


## 4. Research Methods and Index Measurement

This section may be divided by subheadings. It should provide a concise and precise description of the experimental results, their interpretation, as well as the experimental conclusions that can be drawn.

### 4.1. Research Methods

#### 4.1.1. Undesirable-Super-SBM Model

To overcome the inherent limitations of traditional DEA model, Tone proposed SBM model by taking relaxation variables into account. To further distinguish the efficiency of effective decision-making units, Super-SBM model was proposed. The specific model is as follows: assuming that there are n DMUs, each DMU has m inputs and s outputs, and the Super-SBM model under the condition of variable returns to scale is:(1)$$\rho^{*}=min\rho =\frac{1+\frac{1}{m}\sum_{i=1}^{m}s_{i}^{-}/x_{ik}}{1-\frac{1}{s}\sum_{r=1}^{s}s_{i}^{+}/y_{rk}}$$
(2)$$s.t.\overset{n}{\underset{j=1,j\neq k}{\sum }}x_{ij}\lambda_{j}-s_{i}^{-}\leq x_{ik}$$
(3)$$\overset{n}{\underset{j=1,j\neq k}{\sum }}y_{rj}\lambda_{j}+s_{i}^{+}\leq y_{rk}$$
(4)$$\overset{n}{\underset{j=1,j\neq k}{\sum }}\lambda_{i}=1$$
(5)$$\lambda ,s_{i}^{-},s_{i}^{+}\geq 0$$
$$i=1,2,\ldots ,m;\gamma =1,2,\ldots ,q,j=1,2,\ldots ,n\left(j\neq k\right)$$
where $\rho^{*}\;$ is the efficiency value of the kth DMU, m and s are the number of input and output indicators, respectively; $s_{i}^{-}$ and $s_{i}^{+}$ are the slack variables of input and output variables, respectively; $\rho^{*}\;$ ≥ 1 indicates DEA efficiency; $\rho^{*}\;$ < 1 indicates that the decision-making unit does not reach DEA efficiency.

#### 4.1.2. Two-Stage Least Squares Estimation Method

Two-stage least squares (2SLS) is a kind of instrumental variable method, which solves the endogeneity problems caused by missing variables, measurement errors and reverse causality by introducing instrumental variables. First, the endogenous variables and instrumental variables are regressed, and then the estimated values of the endogenous variables obtained from the regression are brought into the original regression equation to form a two-stage least square regression.

#### 4.1.3. Panel Quantile Regression

The quantile regression can capture the extent to which the explained variables are affected at different quantile levels, making the estimation results more robust. To further explore whether the effect of green innovation efficiency on carbon emission is heterogeneous under different carbon emission distribution, this paper describes the whole conditional distribution of carbon emission by quantile regression, and estimates the impact of green innovation efficiency on carbon emission reduction under conditional distribution.

### 4.2. Index Measurement

#### 4.2.1. Carbon Emissions Calculation

Carbon emissions calculation. Carbon dioxide emissions at prefecture-level cities come from the CEADs database. Chen et al. used the characteristics of high correlation between nighttime light data and human activities; two sets of nighttime light data (DMSP/OLS and NPP/VIIRS data) were provided by NGDC (National Geophysical Data Center) to inverse the CO_2_ emissions of 2735 counties in China. In the calibration of nighttime light data, the particle swarm optimization-reverse propagation (PSO-BP) algorithm is adopted to unify the DMSP/OLS and NPP/VIIRS satellite images to obtain high-quality stable nighttime light data within an extended period. The carbon dioxide emissions at the municipal level are derived from the CEADs database, which is aggregated by the carbon dioxide emissions at the county level.

#### 4.2.2. Green Innovation Efficiency

This paper uses the Undesirable-Super-SBM model to measure. In terms of the selection of input indicators, followed by the practice of Wu Chao et al. [34], Xiao Liming et al. [35] and Lv Chengchao et al. [36], this paper measures the input of green innovation activities from three aspects: human resources, financial resources, and energy input. The full-time equivalent of R&D personnel is taken as human input, the internal expenditure of R&D funds is taken as financial input, and the total energy consumption (converted into a standard ton of coal) is taken as energy input. Due to the lag effect of capital investment, the current R&D investment cannot really reflect the actual R&D expenditure of current innovation activities. Therefore, this paper draws on the practice of Lv Yanwei [37] to calculate the stock of R&D expenditure of prefecture-level cities by the method of sustainable inventory as financial input. This paper considers the environmental benefits and economic effects of green innovation output for the expected output. For the output of innovation activities, most scholars choose the number of invention patent applications or authorizations as the expected output. Considering the impact of innovation activities on the ecological environment, this paper uses the amount of authorized green patents to measure the output of innovation activities. For economic benefits, this paper uses the sales revenue of new products to measure the economic benefits of innovation activities. For the unexpected output, considering the impact of innovation activities on resources and the environment, this paper takes the three industrial wastes as the unexpected output into the model, which indirectly reflects the green efficiency. The specific index system is shown in the Table 1:

### 4.3. Data Sources

There are three main sources of data used in this empirical study. The carbon dioxide emissions at the municipal level are derived from the CEADs database, which is aggregated by the carbon dioxide emissions at the county level. The data and control variables involved in the measurement of green innovation efficiency mainly come from the China City Statistical Yearbook, Statistical Yearbook of prefecture-level cities, Statistical Bureau and Science and Technology Bureau of prefecture-level cities

## 5. Carbon Emission of Urban Agglomeration in Yangtze River Delta

### 5.1. Overall Status of Carbon Emissions in Yangtze River Delta Urban Agglomeration

As shown in Figure 1, overall, from 2010 to 2019, the total carbon emissions of the Yangtze River Delta urban agglomeration increased year by year, which is basically consistent with the trend of national carbon emissions change, indicating that the carbon emission reduction policies of the Yangtze River Delta urban agglomeration still have a lot of room for implementation, and need to further promote carbon emission reduction work combined with their own endowment advantages. However, while the proportion of carbon emissions from urban agglomerations in the Yangtze River Delta in China’s carbon emissions has decreased year by year, it decreased from 14.96% in 2010 to 13.13% in 2019. Moreover, the growth rate of carbon emissions in the Yangtze River Delta showed a fluctuating downward trend. It shows that the Yangtze River Delta urban agglomeration has achieved certain results in carbon emission reduction during “Twelfth Five-Year Plan” and “Thirteenth Five-Year Plan”. 

### 5.2. Analysis of Carbon Emissions in Three Provinces and One City of Yangtze River Delta

As shown in Figure 2, in terms of provinces, Jiangsu Province has the highest proportion of total carbon emissions and per capita carbon emissions, far exceeding that of two provinces and one city. It can be seen that the situation faced by Jiangsu Province in energy conservation and emission reduction is not optimistic, and the task of achieving the goal of carbon peak carbon neutralization is still arduous. Shanghai’s carbon emissions accounted for the lowest proportion, about a quarter of Jiangsu’s carbon emissions, and Shanghai’s s total carbon emissions showed a downward trend year by year, indicating that Shanghai’s s carbon reduction work has remarkable results. The proportion of carbon emissions in Anhui Province and Zhejiang Province showed an opposite trend over the years. Total carbon emissions in Anhui Province continued to rise from 2010 to 2014. From 2014 to 2019, the proportion of carbon emissions in Anhui Province slightly decreased year by year. In contrast, the proportion of carbon emissions in Zhejiang Province first decreased and then slightly increased during the study period.

As shown in Figure 3, in terms of per capita carbon emissions, Shanghai’s per capita carbon emissions showed a fluctuating downward trend. From 2010 to 2014, per capita carbon emissions in Anhui Province continued to rise and reached a peak of 6.61 tons per person in 2014, indicating that the per capita output effect of carbon emissions is the main driving force of carbon emissions in Anhui Province. The per capita carbon emission of Zhejiang Province shows an “M”-type fluctuation trend. The per capita carbon emission of Zhejiang Province shows an “M”-type fluctuation trend. From 2014 to 2019, the change trend of per capita carbon emissions in Anhui Province and Zhejiang Province is basically the same, showing a fluctuating upward trend.

As shown in Figure 4, for the carbon emissions per unit GDP, the carbon emissions per unit GDP of three provinces and one city in the Yangtze River Delta continued to decline, indicating that in recent years the Yangtze River Delta region has achieved certain results by adjusting industrial structure and developing low-carbon technologies. Among them, carbon emissions per unit GDP in Anhui Province is the highest, followed by Zhejiang and Jiangsu, and Shanghai is the lowest. It shows that the task of adjusting industrial structure, promoting industrial transformation, and upgrading and eliminating backward production capacity is the most arduous in Anhui Province. 

## 6. Empirical Analysis

### 6.1. Analysis of Benchmark Regression Results

Considering that the data in this paper are balanced panel data, this paper designs a fixed effect model to empirically test the impact of green innovation efficiency improvement on carbon dioxide emissions in various prefecture-level cities in the Yangtze River Delta. The specific measurement model is designed as follows:(6)$$CO2_{it}=\alpha +\beta gie_{it}+\sum x_{k}cv_{k}+\varphi_{t}+\gamma_{i}+\varepsilon_{it}$$
where $CO2_{it}$ represents the carbon dioxide emission of prefecture-level city i  in year t, and its unit is million tons. $\sum x_{k}cv_{k}\;$ represents the set of control variables in this paper. Referring to the existing research, the following variables are included in the control variables: PGDP refers to the per capita output value, which is expressed by dividing the gross domestic product of the region in the current year by the total population; fdi, expressed in terms of actual foreign investment in the current year; pop_ Density stands for population density, which is the ratio of permanent population to total area; produ_aver refers to the urbanization rate, which is expressed by the ratio of resident population to administrative area; third_r represents the proportion of tertiary industry t, which is expressed by the proportion of the output value of tertiary industry in the GDP of the region; Sulfur dioxide, industrial dust emissions are expressed as so2, indu_dust, unit tons; PM2.5 is the concentration of fine particles in micrograms per cubic meter; ti stands for the proportion of science and technology expenditure in the general public budget of the government; Greenland represents the greenery covering area, unit hectare; $\varphi_{t}$ and $\gamma_{i}$ represent the annual fixed effect and regional fixed effect, which are used to the interference of time trend and eliminate the influence of individual characteristics at the level of prefecture-level cities that do not change with time. $\varepsilon_{it}$ is a random disturbance term.

The regression results of the model are shown in Table 2. The first column represents the results of OLS regressions that control years and regions but do not include control variables. The results show that the green innovation efficiency coefficient is negative, and the 1% aboriginality test shows that green innovation efficiency can significantly reduce carbon dioxide emissions in the Yangtze River Delta. The second column indicates the regression results of adding control variables but not considering the year fixed effect and regional fixed effect. The coefficient of green innovation efficiency is still negative. Additionally, the absolute value of the coefficient becomes larger, indicating that the influence of green innovation efficiency on carbon dioxide emissions is amplified by adding control variables. The third column represents the regression results of adding all control variables and controlling the year fixed effect. The fourth column represents the regression results of adding all control variables and controlling the regional fixed effect. The results suggest that the coefficient and significance of green innovation efficiency in the regression results of the regional limited effect model are higher than those of the year fixed-effect model. It shows that the effect of green innovation efficiency on reducing carbon dioxide emissions is more affected by regional changes than by time trends. The reason for this result may be that there is a significant spatial imbalance in the green innovation efficiency of each prefecture-level city in the Yangtze River Delta urban agglomeration due to the influence of location factors and administrative policies [38]. Additionally, the intensity of emission reduction at the urban level is different. During the 13th Five Year Plan period, only 12 prefecture-level cities defined the specific peak time, but there was a lack of clear peak goals and routes [39]. The fifth column is a two-way fixed effect model with all control variables. It shows that the impact of green innovation efficiency on carbon dioxide emissions is still negative after controlling the interference of time and region. Moreover, it passed the test at the level of 10% visibility, indicating that green innovation efficiency can significantly reduce carbon dioxide emissions.

### 6.2. Endogeneity Test

In this paper, the control variables that may affect regional carbon dioxide emissions and the fixed effects of regional and annual interference are added to the empirical model, which can overcome the endogenous problems caused by missing variables to a certain extent. This paper has confirmed that green innovation efficiency can significantly reduce carbon dioxide emissions in the estimation results of each model. However, new technologies and equipment invented by a region to reduce carbon dioxide emissions could boost green innovation efficiency in the region. To overcome the endogeneity problem that may be caused by mutual causality and improve the accuracy of estimation, this paper uses the green innovation efficiency of a province other than itself as the instrumental variable of green innovation efficiency, referring to the method of constructing instrumental variables by Su Danni [40]. The rationality of this approach lies in the following aspects.

On one hand, the green innovation efficiency among different regions within a certain province is closely related, and the provincial government should plan the overall situation in the development process and not cut the innovation links among different regions. It ensures the correlation between instrumental variables and core explanatory variables. On the other hand, carbon dioxide emissions are carried out in the region, and the impact of green innovation efficiency in other regions on carbon dioxide emissions in the region is weak, which ensures the exogeneity between the instrumental variable and the explained variable. Therefore, the tool variable is reasonable.

The results of two-stage least squares estimation (2SLS) using this instrumental variable are shown in the first and second columns of Table 3. In the regression results of the first stage, the instrumental variables passed the test at the significance level of 1%. The second phase of the regression results show that increased green innovation efficiency reduced carbon dioxide emissions and passed the test at the significance level of 5%. The statistics of Cragg Donald Wald F and Kleibergen PAAP Wald F are 21.023 and 23.429, respectively, which are greater than the critical value of 16.38. It can be considered that there is no weak instrumental variable problem at the level of 10% dominance. This further proves that the basic research conclusion of this paper is still stable after solving the endogenous problem. That is, green innovation efficiency can significantly reduce carbon emissions in the Yangtze River Delta region and promote early carbon peak carbon neutralization in the Yangtze River Delta region. It is worth mentioning that the coefficient of green innovation efficiency is higher than those of OLS regression and the fixed effect model in the regression results of 2SLS with instrumental variables. This shows that endogenous problems caused by bidirectional causality lead us to underestimate the effect of green innovation efficiency on reducing carbon dioxide emissions.

### 6.3. Robustness Test

To eliminate the possible influence of extreme value samples on the conclusions of this study, the whole sample is reduced by 5%. That is, we remove the highest 5% sample and the lowest 5% sample of carbon dioxide emission, and then estimate Equation (1). The results are shown in the third column of Table 3. The coefficient of green innovation efficiency is still negative and passes the 5% level of dominance test, which confirms the robustness of the conclusions of this paper.

### 6.4. Heterogeneity Analysis

#### 6.4.1. Regional Heterogeneity

There are great differences in economic development, technological level, environmental conditions, and resource endowments among prefecture-level cities in the Yangtze River Delta. Affected by regional location and administrative policies, there is a significant spatial imbalance in the efficiency of green innovation in various regions. Teng Tangwei et al. found that the green innovation efficiency of urban agglomeration in the Yangtze River Delta was on the rise as a whole, but there was heterogeneity among cities [38]. Therefore, on the basis of the above empirical analysis, this paper uses the two-stage least squares method to group the samples according to the provinces in order to further explore the heterogeneity of the quantitative contribution of green innovation efficiency to carbon peak and carbon neutralization at the provincial level. 

The regression results are shown in Table 4. The regression results show that the coefficient of green innovation efficiency is −0.0034 and does not pass the significance test for Jiangsu Province. Green innovation efficiency in Jiangsu Province can reduce carbon dioxide emissions, but the effect is not obvious. The reason may be that the overall innovation level of Jiangsu Province is high, and the environment is in good condition, which is on the right side of the turning point of the Environmental Kuznets Curve. Therefore, the constraint effect of green innovation efficiency on carbon emissions in Jiangsu Province is weak. For Zhejiang and Anhui Provinces, the coefficients of green innovation efficiency are −0.1536 * and −0.1144 * respectively, which have passed the test at the level of 10% aboriginality and are greater than the regression coefficients of Jiangsu Province. It shows that the effect of green innovation efficiency on reducing carbon dioxide emission is stronger in Anhui and Zhejiang Provinces compared with Jiangsu Province. The possible reason for this is that the industrial intensity effect has inhibited the growth of carbon emissions for most prefecture-level cities in Anhui Province and Zhejiang Province [29]. The improvement of green innovation efficiency can force enterprises to eliminate backward production capacity, and make their industrial structure show a continuous optimization trend, so as to improve environmental quality and contribute to carbon emission reduction.

#### 6.4.2. Heterogeneity under Different Carbon Emissions

Based on the above empirical analysis, to further explore whether the effect of green innovation efficiency on carbon emissions is heterogeneous under different carbon emission distributions, this paper uses the panel quantile model proposed by Powell to describe the overall conditional distribution of carbon emissions through quantile regression and estimates the effect of green innovation efficiency on carbon emission reduction under the conditional distribution. Considering the endogeneity problem, the panel quantile model still adopts the mean value of the green innovation efficiency of a city’s province in addition to itself as a tool variable of green innovation efficiency. Additionally, this paper uses the Markov chain Monte Carlo algorithm to estimate parameters. The estimation results of the panel quantile model are shown in Table 5.

It can be seen from Table 5 that the impact of green innovation efficiency on carbon emissions has obvious heterogeneity among different quantiles. It still supports the basic conclusion that green innovation efficiency can promote carbon emission reduction. In Figure 5, the effect of green innovation efficiency on carbon emissions at different quantiles is the “inverted u-shaped” shape. Under different carbon emission distributions, the impact intensity is different, and the impact intensity in the medium carbon emission cities is greater than that in the lower carbon emission cities and higher carbon emission cities. Specifically, at the 50th, 60th, and 70th quantiles, which correspond to the medium carbon emission cities, the estimated coefficients on gie are −0.347, −0.361 and −0.359 and significant at the 1% level, respectively. It shows that green innovation efficiency has the strongest effect on emission reduction in cities with medium carbon emissions. The reason for this may be that low-carbon urban carbon emissions are relatively low and close to carbon neutralization. However, the reasons for restricting high-carbon-emission cities are complex, which may include resource endowment, development mode and policy reasons. Therefore, the constraint effect of green innovation efficiency on carbon emissions of low-carbon cities and high-carbon cities is relatively weak. The reason may be that the carbon emission level of low-carbon cities is relatively low. Through the adjustment of industrial structure, the original high-energy consuming industries gradually withdraw through relocation and rectification, and the proportion of emerging manufacturing and digital industries in GDP increases [39]. Due to the small space for improving the efficiency of green innovation, the effect of improving the efficiency of green innovation on carbon emission reduction is weak. The reasons that restrict high carbon emission cities are more complex, which may include resource endowment, development mode and policy reasons. Therefore, the efficiency of green innovation has a weak restrictive effect on the carbon emissions of low-carbon-emission cities.

## 7. Research Conclusions and Policy Recommendations

### 7.1. Research Conclusions

Based on the panel data of prefecture-level cities in the Yangtze River Delta from 2010 to 2019, this paper empirically explores the quantitative contribution of green innovation efficiency to carbon peaking and carbon neutralization in the Yangtze River Delta by using OLS regression, fixed effect model and two-stage least square estimation method. The conclusions are as follows: the green innovation efficiency can significantly reduce carbon emissions in the Yangtze River Delta, promote carbon emissions in the Yangtze River Delta to reach the peak as soon as possible, and realize the long-term goal of carbon neutralization as soon as possible. Additionally, this conclusion is still stable after solving the endogenous problem and the influence of outliers. In addition, the effects of green innovation efficiency on carbon emission reduction in Anhui and Zhejiang Provinces are remarkable, and the effect on Zhejiang Province is greater than that of Anhui Province at the provincial level. Moreover, the impact of green innovation efficiency on carbon emissions has obvious heterogeneity between different quantiles, showing an “inverted U” shape, and its intensity in the context of medium carbon emissions is greater than that of low and high carbon emissions.

### 7.2. Policy Recommendations

According to the research conclusion of this paper, improving the efficiency of green innovation is an important measure to reduce carbon dioxide emissions in the Yangtze River Delta region and realize the carbon neutralization of carbon peak at an early date.

(1) In this regard, governments at all levels should improve policies and measures, focus on the adjustment and transformation of traditional industries and the development of new energy, energy conservation and environmental protection and other emerging industries, strengthen the research and development of green innovative technologies, establish and improve the internal incentive mechanism for green technological innovation, and stimulate the endogenous driving force of green innovation.

(2) There is regional heterogeneity in the emission reduction effect of green innovation efficiency, so the regional gap of three provinces and one city in the Yangtze River Delta should be taken into account when formulating the strategic goal of carbon peaking and carbon neutralization. We should strengthen provincial planning, actively guide cities with high green innovation efficiency and cities with low green innovation efficiency to carry out various forms of innovation cooperation and exchange, and form a good inter regional collaborative innovation mechanism. Take effective measures to reduce emissions according to local conditions.

(3) For regions with high carbon emissions, in addition to taking measures to improve the efficiency of green innovation, we should actively develop advanced green and low-carbon environmental protection technologies, adjust the industrial structure and energy structure, promote the transformation and upgrading of industrial structure, formulate practical emission reduction plans, improve management systems, strengthen institutional incentives, and promote the real implementation of the carbon peak goal in combination with our own resource endowment and development reality. For low-carbon emission areas, improving the efficiency of green innovation has little effect on carbon dioxide emission reduction. Therefore, we should speed up the transformation of economic development mode and cultivate new, environment-friendly and sustainable economic growth points.

### 7.3. Limitations and Prospect

There are some deficiencies in this paper: (1) This paper measures the efficiency of green innovation with the help of the unexpected super SBM model. On this basis, this paper combines a fixed effect model, a two-stage least square estimation method and a panel quantile model to carry out research. Although the estimation results are well explained, this article does not take into account the possible spatial effects of variables. (2) The Yangtze River Delta region is increasingly showing a multi-center, networked spatial development phenomenon. However, this paper uses urban panel data and takes cities as independent samples, and has not considered the possible impact of the networked pattern of urban agglomeration in the Yangtze River Delta on variables.

Therefore, in the next study, the possible spatial spillover effects of carbon emissions and green innovation efficiency will be fully considered, and the research will be carried out in combination with the current situation of the multi-center network of the Yangtze River Delta urban agglomeration. We expect to draw more meaningful conclusions in the next study, which will supplement and improve the research of this paper.

## Figures and Tables

**Figure 1 ijerph-19-10245-f001:**
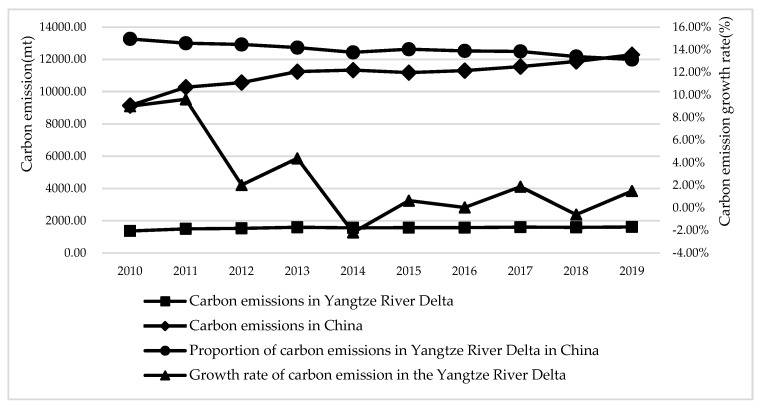
Total carbon emissions and growth rate in the Yangtze River Delta and China.

**Figure 2 ijerph-19-10245-f002:**
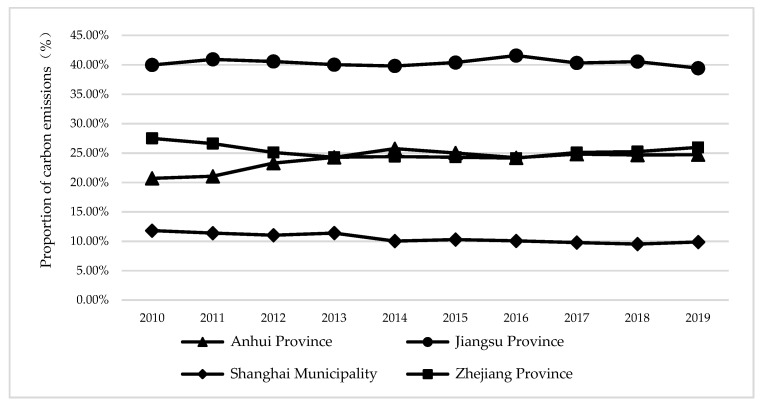
Proportion of carbon emissions from three provinces and one city in the Yangtze River Delta.

**Figure 3 ijerph-19-10245-f003:**
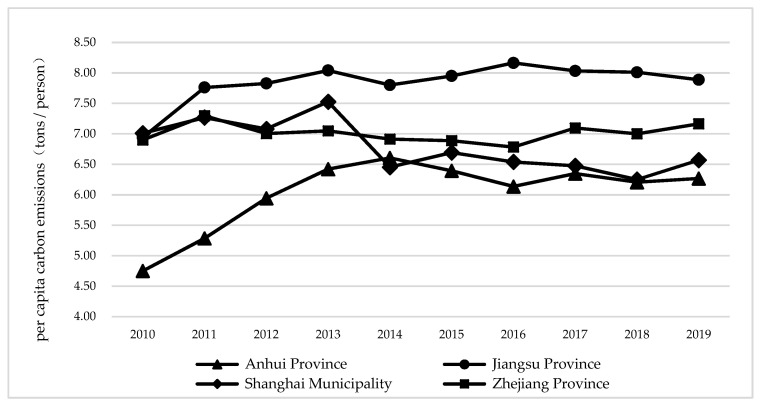
Per capita carbon emission of three cities in the Yangtze River Delta.

**Figure 4 ijerph-19-10245-f004:**
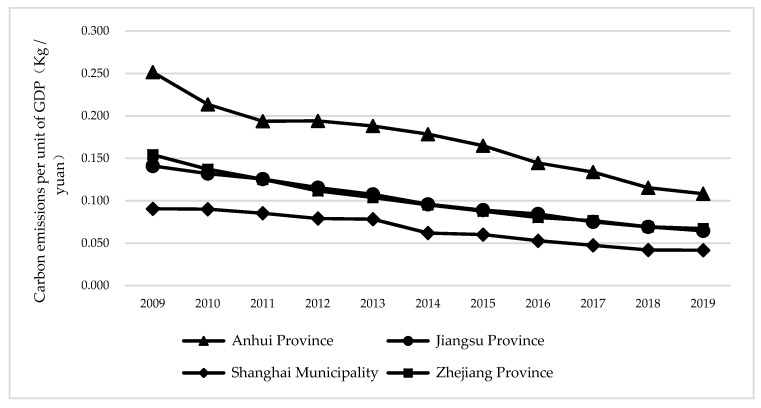
Carbon emission per GDP of three provinces and one city in Yangtze River Delta.

**Figure 5 ijerph-19-10245-f005:**
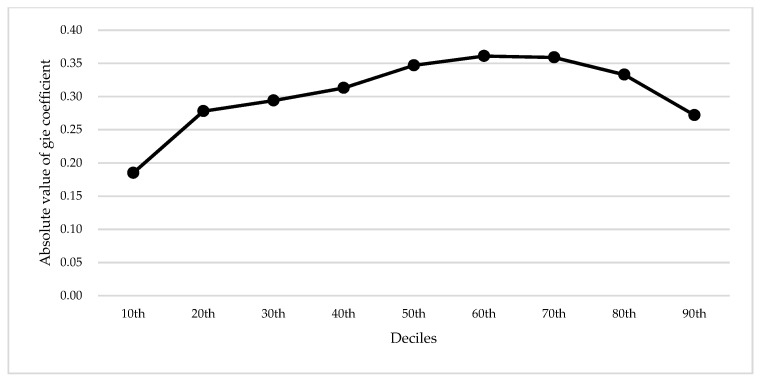
The coefficients of gie in quantile regression.

**Table 1 ijerph-19-10245-t001:** Green innovation efficiency index system.

Primary Index	Secondary Index	Tertiary Indicators	Unit
Input indicators	Human capital	The full-time equivalent of R&D	10,000 People
	Capital investment	R&D capital stock	Hundred billion Chinese yuan
	Energy input	Total industrial energy consumption	10,000 tons of standard coa
Output indicators	Expected outputs	The amount of authorized green patents	Piece
		Sales revenue of new products	Hundred billion Chinese yuan
	Undesired outputs	Industrial waste gas	10,000 tons
		Industrial wastewater discharge	Ton
		Industrial smoke (powder) dust emission	Ton

**Table 2 ijerph-19-10245-t002:** Benchmark regression results.

	(1)	(2)	(3)	(4)	(5)
Variable	I	II	III	IV	V
gie	−0.1811 *** (−3.54)	−0.3169 *** (−4.64)	−0.3158 *** (−5.96)	−0.0170 (0.56)	−0.1341 * (−1.91)
pgdp		0.0007 (0.24)	0.0033 * (1.71)	0.0026 (1.21)	0.0011 (0.73)
fdi		0.0899 ** (2.31)	0.1001 ** (2.02)	−0.2170 *** (−4.96)	−0.1121 ** (−2.25)
pop_density		0.2775 *** (3.86)	0.2110 *** (3.30)	−0.0475 (−0.68)	−0.2479 (−1.13)
produ_aver		2.0783 *** (4.70)	1.7502 ** (2.48)	2.2988 *** (4.18)	1.2084 ** (2.23)
third_r		2.5162 ** (−2.08)	0.6468 (1.39)	1.9042 ** (2.35)	−1.2504 * (−1.74)
so2		−1.36 × 10^−7^ (−1.30)	7.08 × 10^−8^ (0.90)	−1.73 × 10^−7^ (−1.55)	2.12 × 10^−7^ * (1.94)
indu_dust		−1.41 × 10^−8^ (−0.64)	4.65 × 10^−8^ * (1.78)	−4.11 × 10^−8^ * (−1.95)	1.86 × 10^−8^ (1.08)
indu_water		−217.5439 * (−1.88)	217.04 * (1.85)	−526.0929 *** (−3.84)	−269.1283 ** (−2.47)
PM2.5		−0.0007 (−0.47)	0.0043 (1.61)	−0.0143 ** (−2.06)	0.0020 (0.60)
ti		9.0269 *** (5.26)	4.6116 ** (2.43)	5.2227 * (1.97)	2.7080 (1.45)
firm_gs		0.0001 *** (8.23)	0.0002 *** (12.52)	0.0001 * (1.91)	0.0001 ** (2.29)
greenland		0.0458 (0.64)	0.1145 ** (2.42)	0.3065 *** (3.28)	0.0788 (1.22)
year	Yes	No	No	Yes	Yes
area	Yes	No	Yes	No	Yes
obs	400	400	400	400	400
R^2^	0.0028	0.7928	0.7285	0.5229	0.3271

Note: *, **, *** are statistical significance at 10%, 5% and 1%, respectively. The numbers in parentheses represent the value of T.

**Table 3 ijerph-19-10245-t003:** Results of two-stage least squares estimation.

Variable	I	II	III
Instrumental variable	0.4824 *** (3.54)	-	-
gie	-	−0.3729 ** (2.38)	−0.1646 ** (−2.05)
pgdp	−0.0022 (−1.21)	0.0035 (1.05)	0.0018 (0.98)
fdi	0.1250 *** (3.90)	0.0143 (0.11)	−0.0985 ** (−2.48)
pop_density	−0.1219 * (−1.87)	0.3424 *** (3.72)	−0.0105 (−0.04)
produ_aver	0.1927 (0.84)	1.9349 *** (3.99)	0.7337 * (1.70)
third_r	−0.7287 * (−1.81)	2.9064 *** (4.66)	−1.3794 * (−1.95)
so2	−2.74 × 10^−7^ ** (−2.19)	−1.36 × 10^−7^ (−0.39)	2.33 × 10^−7^ ** (2.89)
indu_dust	−7.17 × 10^−9^ (−0.46)	−1.47 × 10^−8^ (−0.59)	2.28 × 10^−8^ (1.27)
indu_water	−296.1486 *** (−3.66)	−217.5439 * (−1.93)	−264.6146 * (−1.94)
pm2.5	−0.0013 (0.10)	−0.0007 (0.43)	−0.0004 (−1.55)
ti	1.167022 (0.99)	9.0269 *** (3.82)	2.67374 (1.50)
firm_gs	0.0001 ** (2.13)	0.0001 *** (4.23)	0.0001 * (1.83)
greenland	−0.1616 *** (−4.47)	0.0458 (1.61)	0.0503 (1.12)
year	Yes	Yes	Yes
area	Yes	Yes	Yes
obs	400	400	340
R^2^	-	0.7462	0.3255

Note: *, **, *** are statistical significance at 10%, 5% and 1%, respectively. The numbers in parentheses represent the value of T.

**Table 4 ijerph-19-10245-t004:** Regional heterogeneity analysis.

	**(** **1** **)**	**(** **2** **)**	**(** **3** **)**	**(** **4** **)**
**Variable**	**Shanghai**	**Jiangsu**	**Zhejiang**	**Anhui**
gie	−0.1811 *** (4.02)	−0.0034 (1.23)	−0.1536 * (−1.89)	−0.1144 * (−1.77)
control variables	control	control	control	control
year	Yes	Yes	Yes	Yes
area	Yes	Yes	Yes	Yes
obs	10	120	110	160
Centered R^2^	0.0028	0.9587	0.9516	0.9538

Note: *, *** are statistical significance at 10%, and 1%, respectively. The numbers in parentheses represent the value of T.

**Table 5 ijerph-19-10245-t005:** Results of panel quantile regression.

**Variable**	**10th**	**20th**	**30th**	**40th**	**50th**	**60th**	**70th**	**80th**	**90th**
gie	−0.185 * (−1.89)	−0.278 *** (−3.03)	−0.294 ** (−2.55)	−0.313 ** (−2.21)	−0.347 *** (−4.29)	−0.361 *** (−3.47)	−0.359 *** (−3.78)	−0.333 *** (−3.84)	−0.272 ** (−2.38)
control variables	control	control	control	control	control	control	control	control	control
year	Yes	Yes	Yes	Yes	Yes	Yes	Yes	Yes	Yes
area	Yes	Yes	Yes	Yes	Yes	Yes	Yes	Yes	Yes
obs	400	400	400	400	400	400	400	400	400
Pseudo R^2^	0.5714	0.5670	0.5564	0.5514	0.5633	0.5839	0.6051	0.6142	0.6298

Note: *, **, *** are statistical significance at 10%, 5% and 1%, respectively. The numbers in parentheses represent the value of T.

## Data Availability

The original data used in this study are mentioned in Section 4 of this paper. The data presented in this study are available on reasonable request from the corresponding author. The data are not publicly available due to privacy restriction.

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
