# Peer review of "Impact of Green Innovation Efficiency on Carbon Peak: Carbon Neutralization under Environmental Governance Constraints"

_ijerph, 2022, doi:10.3390/ijerph191610245_

Round 1

Reviewer 1 Report

Thank you for providing me with the opportunity to review your paper. I have enjoyed reading it. However, I believe that further work is necessary before the paper is suitable for publication.

One major concern relates to the lack of academic strength of the paper. More specifically, it is not very clear what contribution your paper makes to the extant literature.

 My review will follow the format taken by the paper.

1. The introduction has some issues. So, it is recommended that rewrite the introduction with an appropriate content and structure.  Please highlight your contribution and novelty of this manuscript with accuracy. Introduction should also present the structure of the paper.

2. The empirical analysis section is at a fairly general level. Some supplementary literature must be added to compare and contrast the key findings with the existing study. The discussion needs greater engagement with the literature to bring it up to an appropriate level.

3. The conclusion must be based on your results and discussion, but it should also provide the main theoretical and practical contributions of the research Although the authors provide a summary of the existing literature in the field, it remains unclear how exactly they extend this literature. The limitations and future research avenues are also part of the conclusion section.

 4. Recheck the references and their style are according to the journal requirements, and in-text and end-text should be the same and vice versa.

I hope my feedback on this paper will help the authors to improve their work.

Author Response

First, thank you very much for your recognition, evaluation and guidance of our research. Your valuable comments not only make this article more rigorous and serious, but also have guiding value for us to engage in more rigorous academic research in the future. We sincerely thank you!

According to your opinion, we will discuss and revise carefully. I have detailed the modification in the attachment.

We are so sorry to boring you so much trouble because of our carelessness. All the detailed changes are marked in the revised manuscript.

At last, thank you for your review and your comments again. We are looking forward to hearing from you.

Reviewer 2 Report

I am attaching my review of the manuscript as a word file.

Author Response

(The authors gave the same response as above.)

Round 2

Reviewer 2 Report

The manuscript looks fine. I suggest it be published.